# Insight into the Epigenetics of Kaposi’s Sarcoma-Associated Herpesvirus

**DOI:** 10.3390/ijms241914955

**Published:** 2023-10-06

**Authors:** Anusha Srivastava, Ankit Srivastava, Rajnish Kumar Singh

**Affiliations:** 1Institute of Medical Sciences, Banaras Hindu University, Varanasi 221005, Uttar Pradesh, India; 2Faculty of Medical Sciences, Charotar University of Science and Technology, Changa 388421, Gujarat, India

**Keywords:** Kaposi’s sarcoma-associated herpes virus, epigenetics, DNA methylation, histone modification, noncoding RNAs, oncogenesis

## Abstract

Epigenetic reprogramming represents a series of essential events during many cellular processes including oncogenesis. The genome of Kaposi’s sarcoma-associated herpesvirus (KSHV), an oncogenic herpesvirus, is predetermined for a well-orchestrated epigenetic reprogramming once it enters into the host cell. The initial epigenetic reprogramming of the KSHV genome allows restricted expression of encoded genes and helps to hide from host immune recognition. Infection with KSHV is associated with Kaposi’s sarcoma, multicentric Castleman’s disease, KSHV inflammatory cytokine syndrome, and primary effusion lymphoma. The major epigenetic modifications associated with KSHV can be labeled under three broad categories: DNA methylation, histone modifications, and the role of noncoding RNAs. These epigenetic modifications significantly contribute toward the latent–lytic switch of the KSHV lifecycle. This review gives a brief account of the major epigenetic modifications affiliated with the KSHV genome in infected cells and their impact on pathogenesis.

## 1. Introduction

Kaposi’s sarcoma (KS), the most common neoplasm of HIV-infected people, is caused by coinfection of Kaposi’s sarcoma-associated herpes virus (KSHV) [1]. KSHV or human herpes virus 8 (HHV8) belongs to the γ-herpesvirus family and is one of the seven known human oncogenic viruses [2]. In addition to Kaposi’s sarcoma, KSHV infection is strongly associated with multicentric Castleman’s disease (MCD), primary effusion lymphoma (PEL), and KSHV inflammatory cytokine syndrome (KICS) [3].

The genome of KSHV is 165–170 kb linear dsDNA and has complex gene organization, which includes overlapping genes, as well as polycistronic mRNAs [4,5]. The central unique coding region of the KSHV genome is approximately 137 kb and is flanked by 15 kb GC-rich terminal repeats (TR) on both ends [6,7]. The infection proceeds when the virus enters the host cell, and the linear viral DNA undergoes circularization by attaching GGC-rich TRs [8]. This is followed by the attachment of the viral genome with the host genome to form an extrachromosomal viral episome [8,9].

KSHV employs two distinct life cycles consisting of latent and lytic phases; however, it utilizes a common tactic to establish latency in the host cells [10,11]. Escape from host immune response-mediated elimination is a prerequisite for the establishment of lifelong persistent infection [12,13]. KSHV has acquired various strategies to manipulate the epigenetic machinery of the host [14]. KSHV causes the viral episome to form a heterochromatin structure, causing restricted viral gene expression throughout the latency [15,16]. A complex switch maintains the balance between the two phases [11]. This switch can be turned on for lytic replication under hypoxia, under oxidative stress, due to certain chemicals, on account of unbalanced inflammatory cytokines or immunosuppression, or as a result of viral coinfections [17,18]. The latent-to-lytic switch is crucial for viral propagation, its pathogenicity, and maintaining the population of latently infected cells [19]. In the latently infected cells, only a couple of KSHV-encoded genes are expressed, and the products of these expressed genes are crucial to maintain the latency [20]. KSHV-encoded LANA is considered as the master regulator of latency with proven potential to promote oncogenesis by interfering with several cellular pathways [21]. LANA is required for the attachment of the viral genome to the host genome and inactivation of well-established tumor suppressors to promote tumorigenesis [21,22]. LANA interacts with p53 and pRb, and causes suppression of their transcription and transactivation activity. Expression of LANA is associated with increased chromosomal instability, leading to a synonymous increase in micronuclei, multinucleation, and aberrant centrosomes, thus favoring KSHV-mediated pathogenesis and cancer [23]. LANA’s interaction with glycogen synthase kinase 3b (GSK-3b) causes an increase in the level of β-catenin, whereas its interaction with DNA methyltransferase 3a (DNMT3a) causes methylation of cadherin-13 and its downregulation, favoring progression of Kaposi’s sarcoma [24,25,26,27]. Inhibition of TGF-β signaling occurs on account of TbetaRII promoter methylation and deacetylation of histones [28,29]. Viral interleukin-6 encoded by KSHV also contributes to cell proliferation, angiogenesis, and tumor progression [30,31]. KSHV-encoded vCyclin and vFLIP are expressed from the same polycistronic operon and are involved in cell-cycle regulation and inhibition of apoptosis, respectively [32,33]. vCyclin has recently been reported to mediate degradation of HIF1α through the noncanonical lysosomal pathway. This activity of vCyclin has high importance during hypoxic reactivation of KSHV [34]. vFLIP is well known to activate the NF-κB pathway by inhibiting apoptosis, and it can also promote reactivation [35]. Other major transcripts expressed in KSHV-positive cells include vGPCR and KSHV-encoded microRNAs [36]. KSHV-encoded vGPCR is capable of mediating a wide range of signaling cascades through reactive oxygen species (ROS) signaling. It is important to note that vGPCR-induced ROS is reported to modulate the expression of well-known DNA methyl transferases [37]. Additionally, vGPCR acts directly on the MAPK kinase and p38 pathways to promote the expression of VEGF by directly acting on HIF1α [38]. Nevertheless, the KSHV genome is known to encode at least 12 precursors of microRNAs (miRNAs). So far, 25 mature miRNAs have been reported from these 12 precursor miRNAs [39]. The validated targets of KSHV-encoded miRNAs include both host and KSHV genome-encoded transcripts. KSHV-encoded RTA represents a classical target for KSHV-encoded microRNAs miR-K12-7 and miR-K12-9. Targeting RTA using these microRNAs represents a natural mechanism for the maintenance of KSHV latency and inhibition of RTA-mediated reactivation [40].

In addition to DNA methylation, histone modifications also play a determining role in regulating the expression of KSHV-encoded genes, and they are considered a central event during chromatinization of the KSHV genome, especially during the early stage of infection [41]. The most common histone modification of the KSHV genome includes methylation and acetylation of H3 histone [42]. It is important to note that a well-orchestrated reversal event of these modifications is a prerequisite for the reactivation of KSHV from latently infected cells [43]. This is why factors such as hypoxia, reactive oxygen species, and ionizing radiation are among the known factors that can induce KSHV reactivation [44,45]. Additionally, chemical compounds such as 12-O-tetradecanoylphorbol-13-acetate (TPA) and butyric acid can also mediate the global epigenetic changes regularly used for in vitro reactivation of KSHV [46]. In this review, we provide detailed insight into the major epigenetic changes happening on the KSHV genome and their effect on the host epigenome.

## 2. The Interplay between the Genome and Epigenome in KSHV Infections

Cancer initiation and development have been largely linked with congenital or acquired aberrant alterations that occur in the form of mutations, insertions, deletions, and recombinations in the chromosome or copy-number alterations, leading to persistent changes in the phenotype [47]. This represents the classical mechanism of diseases based on the principles of genetics. However, considering the low frequency of genetic events, it cannot be designated as the sole cause of malignant transformations. Epigenetic control has been proposed as an alternate strategy to explain the additional possible mechanisms. Epigenetic changes, being dynamic and causing heritable modifications to the genome, can lead to the accumulation of stable oncogenic traits without causing alterations in DNA sequences, thus contributing to malignancies by exerting a significant effect on the cellular phenotype [48]. Epigenetic modifications during oncogenesis occur mainly through DNA methylation, histone modifications, and noncoding RNAs [49]. Oncogenic viruses represent a group of viruses capable of tumorigenic transformation of infected cells. These viruses range from few kilobase pairs in size such as human papillomavirus (HPV) to more than hundred kilobase pairs in length such as Kaposi’s sarcoma-associated herpesvirus (KSHV) and Epstein–Barr virus (EBV) [50,51]. Both the genomic and the epigenomic aspects of cancer initiation have been proven valid for viral infection-based oncogenesis [52]. The small viruses are often found inserted within the coding regions of important tumor suppressor genes, resulting in a loss of function [53]. Large oncogenic viruses, in general, encode many proteins capable of inducing tumorigenesis [54]. These proteins have potential to activate/stabilize the expression of genes that can promote cell-cycle progression and/or DNA replication or downregulate/degrade tumor suppressors [54]. Homologs of host genes have also been reported in several such viruses, and their infection can represent an external factor mimicking gene duplication/multiplication [55]. Oncogenic viruses also encode proteins which can mediate epigenetic changes through multiple mechanisms [56]. One such example is KSHV-encoded vGPCR, which is capable of large-scale epigenetic changes via reactive oxygen species-dependent expression of DNA methyl transferases [57]. The unerring epigenetic status is a requisite for maintaining and developing tissue-specific gene expression in mammals [58]. Upsetting epigenetic regulation can lead to aberrant gene expression and diseases such as cancer. The disruption of epigenetic status occurs through multiple interconnected mechanisms such as abnormal DNA methylation (e.g., faulty methylation of cytosine residues in CpG sequence motifs and nucleosome remodeling), disrupted pattern of histone post-translational modification, and deregulation by noncoding RNAs (ncRNAs) [59,60]. These modifications result in neoplastic transformation and play a significant role in tumorigenesis. The major epigenetic modifications occurring on the KSHV genome and their impact on KSHV biology are summarized below.

## 3. DNA Methylation

CpG is refers to a cytosine–guanine sequence separated by one phosphate group. These islands are the sites in DNA sequences that are rich in CpG dinucleotides. CpG islands can be found in promoter and exonic regions in approximately 40% of the genes in mammals; however, other regions of the mammalian genome carry very few CpG dinucleotides as these are highly methylated [61]. The human genome is known to possess 42% GC content; accordingly, a nucleotide pair having a cytosine followed by a guanosine can be expected to occur at a rate of 0.21 × 0.21 = 4.41%. However, the observed frequency of these CpG dinucleotides is less than one-fifth of the frequency expected in normal conditions. The reason for this can be attributed to the fact that CpG islands are the genetic hotspot for mutations that lead to CpG depletion during the course of evolution [62]. CpG islands mostly lie within or near gene promoter regions and exhibit lower levels of methylation. Nevertheless, DNA methylation at CPG islands causes loss of optimal chromatin organization and ultimately leads to gene repression, as well as transcriptional impediment. These alterations in the DNA methylome have mostly been observed during carcinogenesis in a wide range of cancer types. Hence, DNA methylation has long been linked to gene silencing, and enough evidence exists to support this paradigm. However, recent studies have shown that promoter gene hypermethylation leads to transcriptional activation, thus making it clear that the above paradigm may not hold true every time. In a review by Smith et al., various instances were reported where DNA methylation acts as a transcriptional activator [63].

The initial instance of methylation at CpG islands in a human tumor suppressor gene was recorded within the retinoblastoma gene in 1989. Following this, in 1994, Herman et al. showed that hypermethylation of unmethylated CpG islands in the VHL gene at the 5′ end could lead to the allelic loss and mutational inactivation of the gene in majority of spontaneous clear-cell renal carcinomas. Methylation of CpG islands in the promoter sequence of genes, particularly the tumor suppressor genes, homeobox genes, and other sequences, results in its failure to transcribe the genes, resulting in silencing of one or more concurrent transcript and, thus, leading to cancer [64,65,66].

In a study conducted in 1983, Gama-Sosa et al. claimed that the overall content of m5C in DNA from normal tissues varies considerably in a manner specific to different tissue. Restriction endonuclease digests of DNA from human tumor samples (secondary malignant, primary malignant, or benign) showed that most of the metastatic neoplasm had significantly lower genomic m5C content than benign or normal tissues [67]. In the same year, Feinberg and Vogelstein observed significant hypomethylation among cancer cells compared to their normal counterparts within four of five patients [68]. These observations concluded that specific genes within the genome of the cancer cells are hypermethylated, but the cumulative mC5 content is still often lowered as a result of cancer-linked hypomethylation or satellite DNA hypomethylation [68]. The above studies curated a view that metastases were more susceptible and, hence, subjected to the cancer-linked DNA hypomethylation influencing epigenetics [69]. Furthermore, there exists tumor-type specificity in cancer-linked hypomethylation [70]. Because of this intricate relationship linking tumor progression, metastasis, and DNA hypomethylation, DNA hypomethylation may represent a promising criterion to classify cancers and predict their clinical status (Figure 1) [71]. Several studies have been reported clearly indicating that the host cellular machinery induces epigenetic reprogramming of the viral genome upon infecting a host cell [72]. In this review, we focus on the epigenetic reprogramming of Kaposi’s sarcoma-associated herpesvirus (KSHV) genome and its role in tumorigenesis.

## 4. DNA Methylation in KSHV Infection

KSHV-infected cells exhibit a distinct cellular gene expression pattern [73]. The genome of KSHV is known to undergo epigenetic modification and chromatinization immediately after entering the host cell. Several DNA methyltransferases (DNMTs) have so far been identified and are responsible for catalyzing DNA methylation. A few examples of such DNMTs include DNMT1, DNMT2, DNMT3, DNMT3A, DNMT3B, DNMT3C, and DNMT3L. However, only some of the DNMTs are known to modulate methylation of KSHV genome [74]. KSHV-encoded factors are also known for their ability to control various aspects of DNA methylation [75]. The main DNA methyltransferase (DNMT3A), encoded by nuclear DNA, can interact with the LANA (latency-associated nuclear antigen or ORF73) encoded by KSHV DNA and is recruited to specific cellular promoters that become methylated and repressed [76].

Ye et al., in 2010, demonstrated the role of DNA methylation in maintaining KSHV latency [77]. It was shown that 5-azacytidine (5-AzaC) which is an inhibitor of DNA methyltransferase acts as a stimulator for KSHV lytic reactivation [14]. Another experiment conducted in 2010 by Gunther and Grundhoff in KSHV-positive endothelial cells showed spatial and temporal DNA methylation and histone modifications associated with the KSHV genome [78]. They employed high-resolution tilling microarrays with immunoprecipitated methylated DNA (MeDIP) and modified histones (ChIP) to work out the distinct landscape of epigenetic modifications that results during KSHV-associated latent infection [78]. They found extensive DNA methylation on the latent KSHV genome except on the latency-associated locus, and it was found that global methylation of the viral episome occurs at a slower rate than histone modification. These findings led to the conclusion that DNA methylation acts as reinforcer of viral gene expression inhibition caused by repressive histone marks [78]. This was followed by a study conducted by Darst et al. in 2013, where they employed MAPit (methylation accessibility probing for individual template) and single-molecule footprinting to map endogenous methylation of CpG islands, accessibility at GC sites and associated chromatin structures at the various loci in the latent KSHV episome [79]. The conclusion drawn from these experiments was in close agreement with that of Gunther and Grundhoff, indicating that DNA methylation can prevent viral reactivation on account of chromatin compaction [79]. Although both Gunther and Grundhoff (2010) and Darst et al. (2013) showed that DNA methylation can significantly contribute to KSHV latency, they also proposed that latency can be developed independently without DNA methylation at the KSHV replication and transcription activator locus, i.e., K-Rta (ORF50).

In a recent study by Journo et al. (2021), the cellular CpG global methylation pattern was observed in KS infected biopsy samples [80]. Methylation EPIC BeadChip was performed to compare the global methylation pattern in normal skin cells and KS biopsy samples, which led to the conclusion that extensive global methylation alterations occur in KS. These alterations can be attributed to the dramatic hypermethylation and hypomethylation of promoters and enhancers of genes that play a role in the regulation of abnormal skin morphology. An inference was made on this basis that hypermethylation occurs early in KS, followed by hypomethylation at a later stage [80].

In many instances, it has been reported that KSHV itself modulates cellular epigenome by utilizing its latent and lytic proteins. One such example is KSHV-encoded vGPCR. KSHV-encoded vGPCR can be transactivated by hypoxia-inducible factor 1 alpha (HIF1α). Under hypoxic conditions, stabilization of KSHV-encoded vGPCR induces production of reactive oxygen species, which in turns modulates the expression of cellular encoded DNMTs. The differential expression of DNMTs in response to vGPCR-mediated reactive oxygen species has been shown to modulate the expression of host nuclear-encoded genes [57].

## 5. Histone Modification

In cells, chromatin serves as a container for DNA. The fundamental component of chromatin is an assembly of histone octamers. The 145–147 base pair DNA fragment is wrapped around a 63 nm central solenoid. These nucleosomes are made up of two copies of each of the four core histone proteins H3, H4, H2A, and H2B, and the linker histone protein H1/H5. The side-chain of the big globular histone proteins is made up of basic lysine and arginine residues [81,82,83]. These histones undergo a number of post-translational covalent changes. Some of these post-translational modifications (PTMs) cause changes in the charge density between the DNA and histones, which affects how chromatin is formed and the associated transcription activities [84]. This can serve as recognition modules for binding of specific proteins that, when bound, may signal chromatin alterations [85]. The modifications can also impact other DNA processes, such as replication, repair, and recombination [86]. Hence, histone modifications are key epigenetic regulators influencing chromatin structure and gene transcription, thus affecting cellular phenotypes [87]. DNA methylation patterns are characterized by mitotic inheritability, which classifies these patterns as epigenetic in more strict sense. However, although histone PTMs are universally labeled as epigenetic, some (e.g., histone acetylation) have a short half-life and are not transmitted autonomously to dividing daughter cells upon cell division [88].

One of the hallmarks of cancer progression is the post-translational modification of histones which can regulate the expression and repression of associated genes [89]. The pioneering work of Vincent Allfrey in 1960 showed that histones are modified post-translationally [90]. The high-resolution X-ray crystal study of nucleosomes by Luger et al. (1997) shed more light on their organization and how histone interactions might cause alterations in chromatin organization [91]. The interpretation was that the highly basic histone amino N-terminal can protrude from the nucleosome. It can interact with nearby nucleosomes, whereby any modification in these tails would result in aberrant inter-nucleosomal interactions, affecting the overall chromatin structure [91]. The modifications can also lead to employment of the remodeling enzymes that can play a role in repositioning nucleosomes by utilizing energy from ATP hydrolysis [92].

Histone modifications can occur as part of histone acetylation, phosphorylation, methylation, demethylation, ADP ribosylation, ubiquitination, or sumoylation [89]. In histone acetylation, there is involvement of histone acetyltransferases (HATs) and histone deacetylases (HDACs) [93]. Utilizing acetyl Co-A as a cofactor, HATs catalyze the transfer of an acetyl group to the ε-amino group of lysine side-chains, neutralizing its positive charge and, thus, weakening the interaction between histones and DNA [93,94]. This weakening leads to chromatin unfolding and exposes charged DNA (negative) to DNA-binding proteins. HDACs are responsible for reversing the action of HATs and restoring the positive charge on lysine, which results in a silencing of gene expression. Transcription dysregulation occurs because of an imbalance between acetylation and deacetylation [95]. Histone phosphorylation predominantly involves serine, threonine, and tyrosine in N-terminal of histone side-chains and alters the charge on histone proteins [96]. Histone methylation involves methylation of side chains of lysine and arginine mediated by histone lysine and arginine methyl transferases, respectively [97]. The key enzymes associated with methylation and demethylation include histone methyltransferase (HMTs) and histone demethylases (HDMs). Enhancer of Zeste homolog 2 (EZH2) is an enzyme which forms the catalytic subunit of polycomb repressive complex 2 (PRC2) and is associated with methylation of lysine 27 in histone H3 (H3K27). Additionally, lysine-specific demethylase 1 (LSD1) performs the function for the removal of the mono/dimethylation marks at lysine 4 and 9 of histone H3 (H3K4/9). It has been observed that H3K9 trimethylation is localized exclusively at repetitive elements or noncoding sequences of genome and in pericentric heterochromatin [98,99]. Until 2002, methylation was considered a relatively stable and static modification. Afterward, Bannister et al. suggested the existence of different potential pathways for demethylation, followed by the discovery of two different classes of lysine demethylase in 2004 and 2006, respectively [85,100]. Histone phosphorylation involves the addition of phosphate moieties at serine, tyrosine, and threonine residues. The addition and removal of these phosphate groups are catalyzed by two enzyme classes, kinases and phosphatases, which contribute to epigenetic up/downregulation [99].

## 6. Histone Modification in KSHV Infection

RTA (replication and transcription activator), encoded by ORF50, acts as Kaposi’s sarcoma-associated herpes virus lytic switch protein [101]. ChIP-on-chip studies demonstrated multiple binding sites for RTA on the KSHV genome in the infected cell line [102]. Two simultaneous but separate chromatin immunoprecipitation (ChIP-on-chip) studies were conducted in 2010, one by Gunther and Grundhoff and the other by Toth et al. [103,104]. Toth et al. conducted an extensive ChIP-on-chip analysis of chromatin of the KSHV genome during latent and lytic phases. The study identified different combinations of activating (H3K4me3 and H3-ac) and repressive histone marks (H3K27me3 and H3K9me3) on the basis of the gene expression class. However, on different positions in KSHV genome, these activating and repressive marks showed a mutually exclusive pattern on bulk of the latent KSHV genome. The colocalization of H3K9me3 with EZH2 histone-lysine N-methyltransferase is responsible for catalyzing histone methylation and transcriptional repression. The role of polycomb repressive complex 2 (PRC2) in bringing about the deposition of H3K27me3 on the KSHV latent genome and, thus, contributing to the KSHV latency was also reported. Viral DNA in latently infected cells has a chromatin structure comprising active and repressive histone marks, unlike the KSHV genome which exists as a chromatin-free structure in virions. The chromatin structure of the viral DNA is influenced by chromatin regulatory factors associated with the KSHV genome during the pre-latency phase of KSHV infection. Toth et al., in 2013, used this observation to explain the biphasic change (biphasic chromatinization) from euchromatin to heterochromatin upon de novo infection [42,105]. Initially, when the infection was initiated (<1 day), euchromatin with elevated levels of active histone marks H3K4me3 and H3K27-Ac were deposited on a viral episome. This was followed by transient induction of a few lytic genes. Post infection (24–48 h), the level of these active marks declined on the KSHV genome, followed by a concomitant increase in the repressive H3K27me3 and H2AK119Ub histone marks, which resulted in dwindling lytic gene expression [42]. This transition was attributed to being dependent on polycomb repressive complexes 1 and 2. The results depicted temporally ordered biphasic euchromatin-to-heterochromatin transition in the case of endothelial cells, resulting in latent infection (Figure 2) [42,105]. On the contrary, the KSHV genome undergoes transcription-active euchromatization in the case of oral epithelial cells, leading to lytic gene expression. These studies concluded that the KSHV genome undergoes differential epigenetic modifications in distinct cell types that govern latent infection and lytic replication of KSHV.

The same conclusion was drawn on the basis of an experiment in which LANA knockout made KSHV incapable of recruiting PRCs to its viral genome [106]. The recruitment of the PRC complex to the viral genome was linked to the genome-wide suppression of lytic gene expression. The factors limiting the expression of lytic genes during the first few hours of infection are an issue highlighted by the discovery of the transient expression of a few lytic genes during the early hours of infection [106]. A probable answer was given in 2017 by Toth et al. on the basis of experiments showing that CTCF and cohesin chromatin organizing factors are recruited before PRCs on the viral genome. Nevertheless, the repression of lytic expression is only due to cohesin, which was labeled as a significant contributor to the persistent latent infection of KSHV in humans [107]. The repertoire of epigenetic factors crucial for establishing and maintaining KSHV latency are vast, and only a few have been deciphered. The role of host epigenetic factors in regulating the complex chromatin structure of KSHV DNA must be decoded to understand viral latency in KSHV pathogenesis. Naik et al., in 2020, carried out an siRNA screen targeting 392 host epigenetic factors during primary infection to see which host epigenetic factors cause the suppression of lytic KSHV genes, eventually depicting their role in establishing latency during primary viral infection [108]. The impact of 392 host epigenetic factors was screened toward primary viral genes responsible for lytic replication (RTA) and latency (LANA) [108]. The group identified the nucleosome remodeling and deacetylase (NuRD) complex, Tip60 and Tip60-associated co-repressors, and the histone demethylase KDM2B posing as inhibitors for the KSHV lytic replication in the latently infected cell, as well as during primary KSHV infection. KDM2B rapidly binds to the viral DNA during the first hours of infection and prevents enrichment of active histone marks on the RTA promoter, leading to the downregulation of RTA expression [108]. This happens before PRCs are recruited on the viral genome. Furthermore, it was found that KDM2B can associate with the viral genome during lytic infection of primary epithelial cells and can suppress viral gene replication and expression, thus positioning KDM2B as a host restriction factor of the lytic cycle during both the latent and the lytic phases of KSHV infection [108].

## 7. Noncoding RNAs

Almost 80% of the eukaryotic genome has been transcribed, but only 2% (mRNA) is considered to have protein-coding function. The remainder of the genome does not code for protein but is transcribed at different levels, and 98% of all transcriptional products were previously considered junk DNA [109,110]. Later studies revealed that noncoding RNAs constitute more than 70% of the human genome [110]. A few of these noncoding RNAs have a putative role in regulating gene expression at the transcriptional and post-transcriptional levels [111]. These noncoding RNAs are segregated into two types according to their function: housekeeping RNAs and regulatory RNAs. The former type encompasses ribosomal RNA (rRNA), transfer RNA (tRNA), small nucleolar RNA (snoRNA), and small nuclear RNA (snRNA), which are expressed constitutively [112]. The latter type with regulatory function includes miRNA, siRNA, piRNA, and lncRNA [112,113].

Additionally, there are covalent closed circular RNAs (circRNAs) which are labeled as the bridge between coding mRNAs and noncoding RNAs. Being expressed abundantly in eukaryotic cells, these RNAs are formed during post-transcriptional processing via a back-splicing mechanism. They act by modulating protein production, RNA transcription, and protein translation, and by sponging the miRNAs. Several studies have been conducted to understand the interaction between virus and host circRNAs, revealing an alteration in the expression of host circRNAs in virus-infected cells. It has been suggested that viruses may use these molecules for their own growth. It has also been observed that certain viruses produce their own viral circRNAs via back-splicing, but the viral genes encoding these circRNAs and their possible function have not been much studied [114,115]. The first noncoding RNA (miRNA) was described in *Caenorhabditis elegans* and was found to be linked to embryogenesis [116]. The relative abundance of non-protein-coding RNAs in eukaryotes is greater than that of protein-coding RNAs [116,117]. Various types of noncoding RNAs involved in influencing epigenetic regulation are (1) small ncRNAs with transcripts shorter than 200 nt, such as siRNA, piRNA, and miRNA, and (2) long noncoding RNAs with transcripts longer than 200 nt, such as lncRNA [118].

siRNA is born out of double-stranded RNA molecules, which can be divided into RNA fragments comprising 19–24 nucleotides using the Dicer enzyme. The fragments exhibit functionality when loaded on Argonaute (AGO) proteins and involve transcriptional gene silencing [118]. Dicer and Argonaute form core components of the eukaryote RNAi machinery [119]. The steps involved in RNAi were interpreted through various in vitro and in vivo experiments. It initiates when RNA nuclease binds to large double-stranded RNA, causing its cleavage into 21–25 nucleotide RNA fragments labeled siRNA. Furthermore, these siRNAs associate with the RNA-induced silencing complex (RISC), leading to homologous single-stranded mRNA degradation [119,120]. piRNAs are the most abundant and diverse ncRNAs, which are approximately 26–31 nucleotides in length. The nomenclature is based on the fact that these ncRNAs interact with Piwi proteins which encode regulatory proteins, giving rise to the PiRNA-induced silencing complex (PiRSC), which contributes to gene silencing and epigenetic reprogramming [121].

MicroRNAs (miRNAs) are single-stranded RNAs comprising 19–24 nucleotides, of which 50% are positioned in chromosomal regions susceptible to structural changes [122]. The mode of action for gene silencing by siRNA and miRNA is speculated to be quite similar due to the similar length of the fragments. However, the two differ in the view that the former is exogenous, originating from a viral infection, the point of gene transfer, or the gene target, while the latter is endogenous, being the expression product of a biological gene. The other notable point of difference is that siRNA is produced from entirely complementary double-stranded RNA. At the same time, the miRNA comprises incomplete hairpin-shaped double-stranded RNA, whereby Drosha and Dicer process the former, but the latter is processed by Dicer only [122,123]. The number of putative microRNAs identified in the human genome is increasing quickly due to the development of sophisticated sequencing techniques such as next-generation sequencing (NGS). Thus, its role in regulating epigenetics is continuously being revealed [124].

LncRNA lacks a protein-coding function, comprises transcripts longer than 200 nucleotides, and shares certain standard features with mRNAs such as 5′-methylguanosine capping, polyadenylation, and splicing [125]. Structurally, lncRNA is less conserved than mRNA [125]. Nevertheless, they exhibit complex secondary structures after interaction with DNA, RNA, and proteins to form a tertiary structure for executing their functional activities [125,126]. lncRNAs regulate diverse biological and cellular processes such as transcriptional and post-transcriptional processing, chromatin remodeling, metabolism, development, and differentiation [127]. Any misregulation gives rise to diseased conditions in humans, including cancer. lncRNAs are known to influence various phenotypes of cancer cells, including proliferation, chemoresistance, and metastasis, and they can pose as potential biomarkers for cancer diagnosis and as a target for treatment [127,128].

## 8. Role of Noncoding RNAs in KSHV Biology

The expression of an array of viral miRNAs was noted in latently infected cells of KSHV by Cai et al. in 2005 [129]. Grundhoff et al., in 2006, employed computational and microarray-based approaches to identify novel miRNAs encoded by KSHV. In addition to the 10 known pre-miRNAs of KSHV, one was added to the list after confirmation of the candidate miRNAs through Northern blot analysis [130]. Following this, Umbach et al., in 2010, provided insight into miRNA processing in mammalian cells, and the data indicated that the process is highly conserved during animal evolution [131]. During latent infection, KSHV expresses 12 pre-miRNAs that can be processed to give 25 mature miRNAs [39,132]. Schifano et al., in 2017, penned 16 potential KSHV-encoded lncRNAs using various experimental approaches [133]. The best-studied species among the known KSHV-encoded lncRNAs are polyadenylated nuclear RNAs (PAN RNAs), described in 1996 [134,135].

Cai et al., in 2004, suggested that, during the initial phase, miRNAs are transcribed as a largely unstructured precursor [136]. This forms part of one arm of the stem loop, constituting part of a long, capped polyadenylated RNA, called primary miRNA (pri-miRNA) [136,137]. This pri-miRNA undergoes nuclear processing by RNAase III enzyme Drosha and the RNA-binding cofactor DGCR8, which results in the cleavage of the pri-miRNA into approximately 65 nt pre-miRNA hairpin intermediates [138,139]. Drosha executes a staggered cut characteristic of RNAase III endonuclease that results in a 5′-phosphate and ~2 nt 3′-overhang. The pre-miRNA is then transported from the nuclear compartment to the cytoplasm by karyopherin family member Exportin 5 [140,141]. After being recognized by another RNAase III member, Dicer, the pre-miRNA undergoes cleavage that removes the terminal loop and forms an intermediate miRNA duplex [142]. One strand of this miRNA duplex is selectively picked up by and incorporated into an RNA-induced silencing complex (RISC). This miRNA then guides the RISC to complementary RNA sequences. If the miRNA–RISC complex locates an RNA sequence with high complementarity, it leads to the cleavage of mRNA due to the activation of RNAase. On the other hand, if the miRNA–RISC complex locates an RNA sequence with imperfect complementarity, it leads to translational repression (Figure 3) [143,144,145].

Chromatin isolation by RNA purification (ChiRP) assay was employed to investigate the role of PAN RNA in regulating viral latency by Rossetto et al., in 2013, who found that PAN RNA was present at multiple sites on the KSHV genome [146]. K-RTA promoter regions bind and facilitate the recruitment of cellular factors, including UTX and JMJD3, which are H3K27me3 demethylases, along with H3K4me3 methyltransferase MLL2 [147]. This binding results in an increase in activation mark H3K4me3 in the K-RTA promoter region and a subsequent decrease in repressive mark H3K27me3, which leads to disruption of viral latency [147]. Association of PAN RNA with ORF59, the DNA polymerase processivity factor of KSHV, might also contribute to the functioning of PAN RNA in activation of the gene expression during the lytic cycle (Figure 4) [148]. However, contrary to this, Rossetto et al., by utilizing ChiRP assays, reported PAN RNA’s presence on KSHV genome and its association with PRC2 components SUZ12 and EZH2. This association resulted in an increase in repressive mark H3K27me3, leading to the repression of gene expression and latency establishment (Figure 5) [149]. Furthermore, additional viral factors that interact with PAN RNA were also examined. According to Campbell et al., one such factor is LANA. PAN RNA dissociates LANA from the viral genome and disrupts KSHV latency. Following this, Withers et al., in 2018, showed that KSHV-encoded PAN RNA, although nuclear, was not associated with chromatin. They utilized capture hybridization analysis of RNA targets (CHART) and nuclear fractionation studies. The results favored chromatin-independent PAN RNA activities. The contrasting features put up by the researchers suggest that PAN RNA’s role is obscure in the viral life cycle [14,150]. The prominent epigenetic signatures by noncoding RNAs on the KSHV genome and in maintaining persistent infection have been well studied. However, more recently, noncoding circRNAs were identified in gamma herpes viruses. In a study be Ungerleider et al., circRNAs derived from the vIRF4 gene and lytic MHV68 circRNAs were identified from the analysis of KSHV’s circRNAs [151].

## 9. Clinical Applications

The morphological manifestation of Kaposi’s sarcoma is in the form of dark-brown macules, purpura, or spots on the skin that lead to ulceration and bleeding. For patients, who are untreated for AIDS, the pace of these spreading to the rest of the body is high and occurs within months. Currently, radiation-induced therapies are employed to treat soft-tissue sarcomas [31]. Epigenetic modifications play a significant role in the development and progression of Kaposi’s sarcoma (KS) associated with Kaposi’s sarcoma-associated herpesvirus (KSHV) infection. However, the clinical application of these modifications and the development of targeted therapies specifically aimed at modulating epigenetic dysregulation in KS are still in their infancy. Although the understanding of epigenetic alterations in KSHV-mediated KS has improved, translating this knowledge into effective clinical interventions remains a challenge. Sun et al. examined AcH3 and H3K27me3 histone modifications on the KSHV genome, as well as the genome-wide binding sites of latency associated nuclear antigen (LANA). They further reported that the enriched AcH3 was mainly restricted at the latent locus, while H3K27me3 was widespread on the KSHV genome in classic KS tissues. Nevertheless, recent advancements in epigenetic research and therapeutic strategies have provided hope for the future. Several ongoing clinical trials are exploring the potential of epigenetic-targeted therapies for KS treatment. These trials aim to evaluate the efficacy and safety of drugs that can specifically modify epigenetic marks, such as DNA methylation and histone modifications, in KSHV-infected cells. Histone demethylase inhibitors may be an option for the treatment of oncogenic viruses such as KSHV and Epstein–Barr virus (EBV), a major cause of Hodgkin’s lymphoma, which seem to depend heavily on histone deacetylases (HDACs) to maintain their latency. Vorinostat, everolimus, and sirolimus were proven in a recent trial to synergistically suppress relapsed, refractory Hodgkin’s lymphoma in patients [152]. Further research and clinical trials are needed to validate the efficacy of these interventions and determine their long-term benefits for patients with this complex and challenging malignancy.

## 10. Conclusions

Unraveling the epigenetic landscape of Kaposi’s sarcoma-associated herpes virus is complex yet compelling. It is important to visualize the underlying molecular interplay and orchestration of various cellular mechanisms involving replication, transcription, translation, and gene expression. Epigenetic factors are instrumental in regulating KSHV’s latent-to-lytic switch that drives disease progression. It is an intriguing field of research that can pave the route for the development of targeted tumor therapies. Significant work has been performed in this line, but the therapeutical aspect has not been much explored. Epigenetic targeted therapy and treatments are still in their early phase and carry great potential for future research.

## Figures and Tables

**Figure 1 ijms-24-14955-f001:**
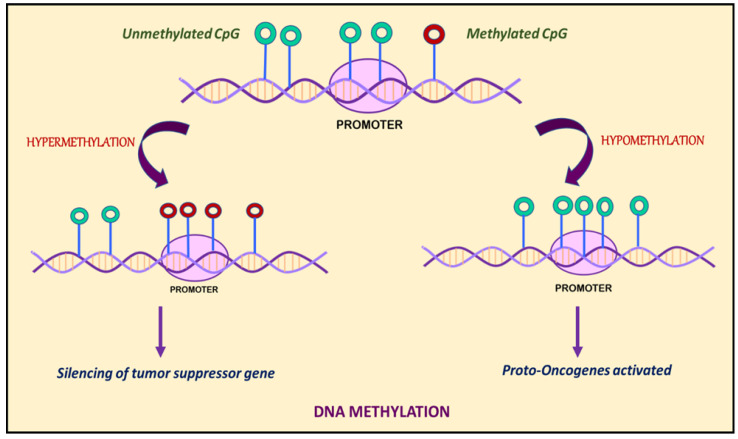
DNA methylation is marked by hypermethylation and hypomethylation of CpG islands. Hypermethylation of promoters or enhancers leads to silencing of tumor suppressor genes, whereas hypomethylation leads to activation of proto-oncogenes.

**Figure 2 ijms-24-14955-f002:**
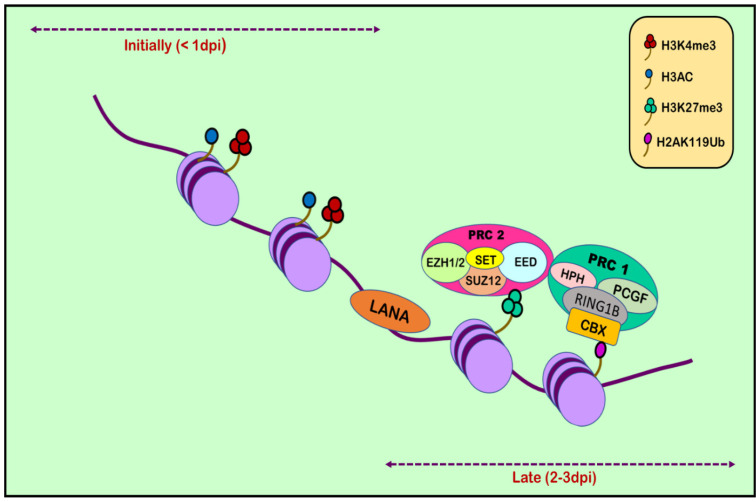
Histone modification in KSHV infection. Upon initial de novo infection (1 dpi), active marks are deposited on the viral episome. Post infection (2–3 dpi), the number of active marks declines, followed by a subsequent increase in repressive marks through the recruitment of polycomb repressive complexes 1 and 2.

**Figure 3 ijms-24-14955-f003:**
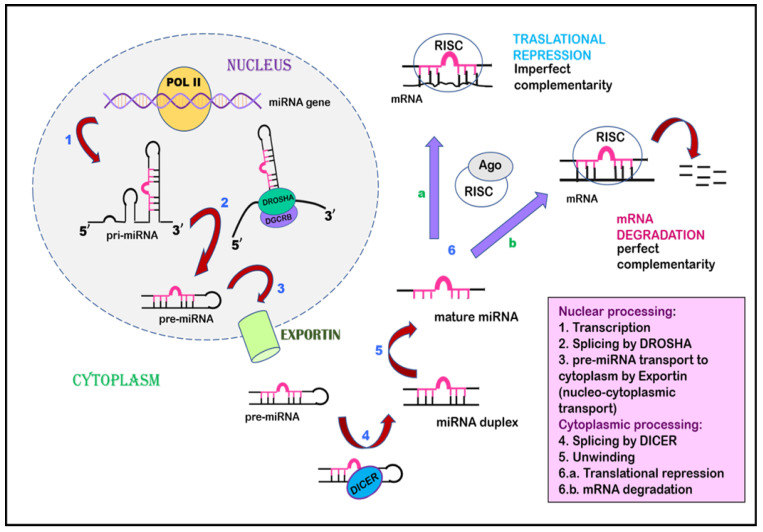
Inhibition of gene expression by miRNA. miRNA undergoes transcription to form primary miRNA, is spliced by Drosha to form pre-miRNA, and is exported from the nucleus to the cytoplasm by Exportin via nucleocytoplasmic transport. The pre-miRNA undergoes splicing by Dicer in the cytoplasm, leading to the formation of an miRNA duplex, followed by mature miRNA. This mature miRNA is incorporated by a multiprotein complex named the RNA-induced silencing complex, which acts as a template to interact with mRNA. This interaction can proceed in two ways: (a) translational repression occurs when miRNA–RISC interacts with mRNA of imperfect complementarity. (b) when miRNA–RISC finds complementary mRNA, RNAse is activated, leading to mRNA degradation.

**Figure 4 ijms-24-14955-f004:**
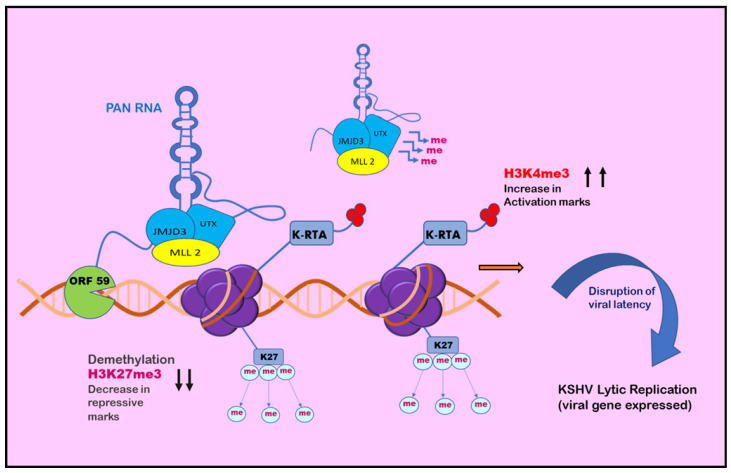
Role of PAN RNA in regulating KSHV latency: positive epigenetic regulation. PAN RNA is found at multiple sites on the KSHV genome. It may bind and recruit cellular factors on regions such as the K-RTA promoter region and may also associate with ORF59. The cellular factors facilitate the decrease in H3K27me3 repressive marks and subsequent increase in H3K4me3 activation marks, leading to disruption of viral latency and initiation of KSHV lytic replication.

**Figure 5 ijms-24-14955-f005:**
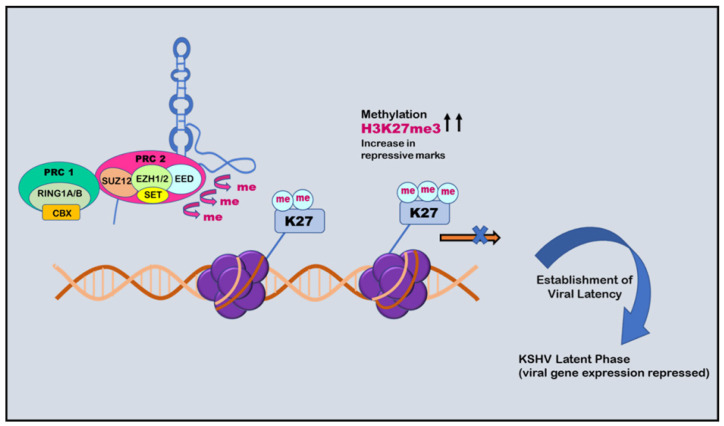
Role of PAN RNA in regulating KSHV latency: negative epigenetic regulation. PAN RNA can recruit polycomb repressive complex 2 (PRC2) that increases H3K27me3 repressive marks and causes methylation. This phenomenon causes repression of the expression of viral genes and establishment of viral latency.

## Data Availability

Not applicable.

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
