# Peer review of "Insight into the Epigenetics of Kaposi’s Sarcoma-Associated Herpesvirus"

_ijms, 2023, doi:10.3390/ijms241914955_

Round 1
Reviewer 1 Report
The Authors have presented a comprehensive review on “Insight into the Epigenetics of Kaposi's Sarcoma Associated Herpesvirus”. In this review authors have presented very precise latest information with schematic diagrams on epigenetic regulation of KSHV infection. Authors have also distributed their review in different sections to better understand its epigenetics regulations. Though authors tried to compile good sets of information, I feel the language of this manuscript is not up to the mark and authors need to take care of the punctuation and English language throughout the manuscript. Authors should also need to take care of the journal's guidelines for citing the work. e.g. In Section of DNA Methylation: in 1994, J G Herman et al. showed that... (Remove J G), In a study conducted in 1983, Gama-Sosa et. Al., (Change to al.)
The authors need to take care of the punctuation and English language throughout the manuscript.
Author Response
Reviewer 1
Comments and Suggestions for Authors
The Authors have presented a comprehensive review on “Insight into the Epigenetics of Kaposi's Sarcoma Associated Herpesvirus”. In this review authors have presented very precise latest information with schematic diagrams on epigenetic regulation of KSHV infection. Authors have also distributed their review in different sections to better understand its epigenetics regulations. Though authors tried to compile good sets of information, I feel the language of this manuscript is not up to the mark and authors need to take care of the punctuation and English language throughout the manuscript. Authors should also need to take care of the journal's guidelines for citing the work. e.g. In Section of DNA Methylation: in 1994, J G Herman et al. showed that... (Remove J G), In a study conducted in 1983, Gama-Sosa et. Al., (Change to al.)
Response: We thank the reviewer for the time and effort in detailed review of our review article. Thank you very much for appreciation and we are happy to know that reviewer finds our review as comprehensive and precise. In the revised version of the review, we addressed the concerns related with language throughout the manuscript. Further, care has been taken for the citing others works according to the journal’s guidelines. Changes has been made for highlighted as well as other references.
Comments on the Quality of English Language
The authors need to take care of the punctuation and English language throughout the manuscript.
Response: Thank you. Proper care has been taken for the punctuation and English language throughout the manuscript.
Reviewer 2 Report
The manuscript entitled “Insight into the Epigenetics of Kaposi's Sarcoma Associated Herpesvirus” (journal: IJMS MDPI) by Dr. Srivastava and colleagues is a review which extensively describes and discuss the most recent findings on the epigenetic mechanisms on Kaposi’s sarcoma-associated herpesvirus genome relying on the onset and development of Kaposi's Sarcoma Associated Herpesvirus. It provides an overview of the epigenetic mechanisms, such as DNA methylation, histone modifications and non-coding RNAs as well as the interplay between Herpesvirus and these epigenetics modifications associated with Kaposi’s Sarcoma occurrence. The review is well written (please see below for suggestions for necessary improvments). The figures are highly informative. However, considering the necessary improvements, I recommend a major revision. I also have several minor observations for improving the work:
I have three major comments:
1 - Epigenetic modifications in the context of Kaposi’s sarcoma-associated herpesvirus and Kaposi's Sarcoma are well described. However, there is neither information on the clinical application of these modifications, nor the current clinical works/clinical trials developed in order to counteract these modifications for Kaposi's Sarcoma therapy. For completeness, I recommend including a paragraph exploring these important aspects, which are as important as the general description of the epigenetics dysregulation and their role in the Kaposi's Sarcoma initiation. In other words, can epigenetic modifications occurring on Kaposi’s sarcoma-associated herpesvirus be targeted for tumor therapy? I recommend this important review manuscript in the field https://www.ncbi.nlm.nih.gov/pmc/articles/PMC8745302/
2 - A conclusive paragraph summarizing the main aspects of the review, which might include some future prospective, is necessary for a better reading of the work and should be included at the end of the manuscript.
3- Given numerous similar works in the field, the novelty of the work should be underlined before the aim.
Please see below some minor suggestion:
1 – Please carefully revise the work for the presence of minor typo errors. For instance, all references are linked to the last word of the sentence, while a space between words should be included.
2- these important recently published references in the field should be included:
https://www.mdpi.com/1422-0067/23/1/422
https://www.frontiersin.org/articles/10.3389/fmicb.2020.00850/full
https://www.ncbi.nlm.nih.gov/pmc/articles/PMC7174275/
https://pubmed.ncbi.nlm.nih.gov/36358814/
3 – section 3, promoter gene hypermethylation in some cases can favor gene expression (PMID:32348735)
4 – section 3 - Please uniform the acronyms. Authors previously mentioned DNMTs, then mentioned “DNA Methyltransferase”
5 - Section 4, “Histone modifications can occur as part of histone, acetylation, phosphorylation,
methylation, demethylation, ADP ribosylation, and Ubiquitination or Sumoylation “ this additional supporting reference on histone modifications should be included (https://pubmed.ncbi.nlm.nih.gov/35350569/)
6 - section 7. among non-coding RNAs there is no mention of circular RNAs
English is fine
Author Response
Reviewer 2
Comments and Suggestions for Authors
The manuscript entitled “Insight into the Epigenetics of Kaposi's Sarcoma Associated Herpesvirus” (journal: IJMS MDPI) by Dr. Srivastava and colleagues is a review which extensively describes and discuss the most recent findings on the epigenetic mechanisms on Kaposi’s sarcoma-associated herpesvirus genome relying on the onset and development of Kaposi's Sarcoma Associated Herpesvirus. It provides an overview of the epigenetic mechanisms, such as DNA methylation, histone modifications and non-coding RNAs as well as the interplay between Herpesvirus and these epigenetics modifications associated with Kaposi’s Sarcoma occurrence. The review is well written (please see below for suggestions for necessary improvements). The figures are highly informative. However, considering the necessary improvements, I recommend a major revision. I also have several minor observations for improving the work:
Response: Than you very much for thorough review of our manuscript, appreciation for the work and providing valuable suggestions. We are happy to know that reviewer finds that the review is well written and extensively describes the most recent advances in the field. We incorporated the suggested changes in the revised manuscript and we hope that the changes will be satisfactory.
I have three major comments:
1 - Epigenetic modifications in the context of Kaposi’s sarcoma-associated herpesvirus and Kaposi's Sarcoma are well described. However, there is neither information on the clinical application of these modifications, nor the current clinical works/clinical trials developed in order to counteract these modifications for Kaposi's Sarcoma therapy. For completeness, I recommend including a paragraph exploring these important aspects, which are as important as the general description of the epigenetics dysregulation and their role in the Kaposi's Sarcoma initiation. In other words, can epigenetic modifications occurring on Kaposi’s sarcoma-associated herpesvirus be targeted for tumor therapy? I recommend this important review manuscript in the field https://www.ncbi.nlm.nih.gov/pmc/articles/PMC8745302/.
Response: Thank you very much for this important comment. We completely agree with the reviewer that available literature lacks information on the clinical applications of these modifications or strategies to counteract these modifications for targeting KSHV-associated diseases. In the revised manuscript, we are added these aspects. Further, as per reviewer’s suggestion, we believe that of incorporating this aspect adds high value to the manuscript. Therefore, we added a separate paragraph exploring these important aspects. The recommended article is now documented in the revised manuscript. We hope that that changes will be satisfactory.
2 - A conclusive paragraph summarizing the main aspects of the review, which might include some future prospective, is necessary for a better reading of the work and should be included at the end of the manuscript.
Response: Thank you for the comment. In the revised manuscript, a conclusive paragraph is added which summarizes the main aspects of the review. Further, future perspectives are also added in the revised manuscript at the proper section.
3- Given numerous similar works in the field, the novelty of the work should be underlined before the aim.
Response: Thank you very much for this important comment. In the revised manuscript, changes have been made according to the reviewer comment. We hope these changes are satisfactory.
Please see below some minor suggestion:
1 – Please carefully revise the work for the presence of minor typo errors. For instance, all references are linked to the last word of the sentence, while a space between words should be included.
Response: Thank you. We carefully revised the manuscript for the presence of any minor typological errors. Also, we rectified any error associated with referencing or spacing between words.
2- these important recently published references in the field should be included:
https://www.mdpi.com/1422-0067/23/1/422
https://www.frontiersin.org/articles/10.3389/fmicb.2020.00850/full
https://www.ncbi.nlm.nih.gov/pmc/articles/PMC7174275/
https://pubmed.ncbi.nlm.nih.gov/36358814/
Response: Thank you. All the above suggested publications are now cited in the revised manuscript at the proper sections.
3 – section 3, promoter gene hypermethylation in some cases can favor gene expression (PMID:32348735).
Response: Thank you. We incorporated this aspect in the revised manuscript and incorporated the suggested reference at adequate section.
4 – section 3 - Please uniform the acronyms. Authors previously mentioned DNMTs, then mentioned “DNA Methyltransferase”.
Response: Thank you for the comment. In the revised manuscript, we used uniform acronyms.
5 - Section 4, “Histone modifications can occur as part of histone, acetylation, phosphorylation,
methylation, demethylation, ADP ribosylation, and Ubiquitination or Sumoylation “this additional supporting reference on histone modifications should be included (https://pubmed.ncbi.nlm.nih.gov/35350569/).
Response: Thank you for the comment. The above-mentioned reference on histone modifications is now included in the revised manuscript.
6 - section 7. among non-coding RNAs there is no mention of circular RNAs.
Response: Thank you for the comment. In the non-coding RNNAs section, we included about circular RNAs. We hope these additions will be satisfactory.
Comments on the Quality of English Language
English is fine
Response: Thank you. Further improvements have been made for the English language.
Round 2
Reviewer 2 Report
The ms can be accepted in the present form